# THE EFFICACY OF $L_1$ REGULARIZATION IN NEURAL NETWORKS

## ABSTRACT

A crucial problem in neural networks is to select the most appropriate number of hidden neurons and obtain tight statistical risk bounds. In this work, we present a new perspective towards the bias-variance tradeoff in neural networks. As an alternative to selecting the number of neurons, we theoretically show that $L_1$ regularization can control the generalization error and sparsify the input dimension. In particular, with an appropriate $L_1$ regularization on the output layer, the network can produce a statistical risk that is near minimax optimal. Moreover, an appropriate $L_1$ regularization on the input layer leads to a risk bound that does not involve the input data dimension. Our analysis is based on a new amalgamation of dimension-based and norm-based complexity analysis to bound the generalization error. A consequent observation from our results is that an excessively large number of neurons do not necessarily inflate generalization errors under a suitable regularization.

## 1  INTRODUCTION

Neural networks have been successfully applied in modeling nonlinear regression functions in various domains of applications. A critical evaluation metric for a predictive learning model is to measure its statistical risk bound. For example, the $L_1$ or $L_2$ risks of typical parametric models such as linear regressions are at the order of $(d/n)^{1/2}$ for small $d$ (Seber & Lee, 2012), where $d$ and $n$ denote respectively the input dimension and number of observations. Obtaining the risk bound for a nonparametric regression model such as neural networks is highly nontrivial. It involves an approximation error (or bias) term as well as a generalization error (or variance) term. The standard analysis of generalization error bounds may not be sufficient to describe the overall predictive performance of a model class unless the data is assumed to be generated from it. For the model class of two-layer feedforward networks and a rather general data-generating process, Barron (1993; 1994) proved an approximation error bound of $O(r^{-1/2})$ where $r$ denotes the number of neurons. The author further developed a statistical risk error bound of $O((d/n)^{1/4})$, which is the tightest statistical risk bound for the class of two-layer neural networks up to the authors' knowledge (for $d < n$). This risk bound is based on an optimal bias-variance tradeoff involving an deliberate choice of $r$. Note that the risk is at a convergence rate much slower than the classical parametric rate. We will tackle the same problem from a different perspective, and obtain a much tighter risk bound.

A practical challenge closely related to statistical risks is to select the most appropriate neural network architecture for a particular data domain (Ding et al., 2018). For two-layer neural networks, this is equivalent to selecting the number of hidden neurons $r$. While a small $r$ tends to underfit, researchers have observed that the network is not overfitting even for moderately large $r$. Nevertheless, recent research has also shown that an overly large $r$ (e.g., when $r > n$) does cause overfitting with high probability (Zhang et al., 2016). It can be shown under some non-degeneracy conditions that a two-layer neural network with more than $n$ hidden neurons can perfectly fit $n$ arbitrary data, even in the presence of noise, which inevitably leads to overfitting. A theoretical choice of $r$ suggested by the asymptotic analysis in (Barron, 1994) is at the order of $(n/d)^{1/2}$, and a practical choice of $r$ is often from cross-validation with an appropriate splitting ratio (Ding et al., 2018). An alternative perspective that we advocate is to learn from a single neural network with sufficiently many neurons and an appropriate $L_1$ regularization on the neuron coefficients, instead of performing a selection from multiple candidate neural models. A potential benefit of this approach is easier hardware

implementation and computation since we do not need to implement multiple models separately. Perhaps more importantly, this perspective of training enables much tighter risk bounds, as we will demonstrate. In this work, we focus on the model class of two-layer feedforward neural networks.

Our main contributions are summarized below. First, we prove that $L_1$ regularization on the coefficients of the output layer can produce a risk bound $O((d/n)^{1/2})$ (up to a logarithmic factor) under the $L_1$ training loss, which approaches the minimax optimal rate. Such a rate has not been established under the $L_2$ training loss so far. The result indicates a potential benefit of using $L_1$ regularization for training a neural network, instead of selecting from a number of neurons. Additionally, a key ingredient of our result is a unique amalgamation of dimension-based and norm-based risk analysis, which may be interesting on its own right. The technique leads to an interesting observation that an excessively large $r$ can reduce approximation error while not increasing generalization error under $L_1$ regularizations. This implies that an explicit regularization can eliminate overfitting even when the specified number of neurons is enormous. Moreover, we prove that the $L_1$ regularization on the input layer can induce sparsity by producing a risk bound that does not involve $d$, where $d$ may be much larger compared with the true number of significant variables.

**Related work on neural network analysis**. Despite the practical success of neural networks, a systematic understanding of their theoretical limit remains an ongoing challenge and has motivated research from various perspectives. Cybenko (1989) showed that any continuous function could be approximated arbitrarily well by a two-layer perceptron with sigmoid activation functions. Barron (1993; 1994) established an approximation error bound of using two-layer neural networks to fit arbitrary smooth functions and their statistical risk bounds. A dimension-free Rademacher complexity for deep ReLU neural networks was recently developed (Golowich et al., 2017; Barron & Klusowski, 2019). Based on a contraction lemma, a series of norm-based complexities and their corresponding generalization errors are developed (Neyshabur et al., 2015, and the references therein). Another perspective is to assume that the data are generated by a neural network and convert its parameter estimation into a tensor decomposition problem through the score function of the known or estimated input distribution (Anandkumar et al., 2014; Janzamin et al., 2015; Ge et al., 2017; Mondelli & Montanari, 2018). Also, tight error bounds have been established recently by assuming that neural networks of parsimonious structures generate the data. In this direction, Schmidt-Hieber (2017) proved that specific deep neural networks with few non-zero network parameters can achieve minimax rates of convergence. Bauer & Kohler (2019) developed an error bound that is free from the input dimension, by assuming a generalized hierarchical interaction model.

**Related work on $L_1$ regularization**. The use of $L_1$ regularization has been widely studied in linear regression problems (Hastie et al., 2009, Chapter 3). The use of $L_1$ regularization for training neural networks has been recently advocated in deep learning practice. A prominent use of $L_1$ regularization was to empirically sparsify weight coefficients and thus compress a network that requires intensive memory usage (Cheng et al., 2017). The extension of $L_1$ regularization to group-$L_1$ regularization (Yuan & Lin, 2006) has also been extensively used in learning various neural networks (Han et al., 2015; Zhao et al., 2015; Wen et al., 2016; Scardapane et al., 2017). Despite the above practice, the efficacy of $L_1$ regularization in neural networks deserves more theoretical study. In the context of two-layer neural networks, we will show that the $L_1$ regularizations in the output and input layers play two different roles: the former for reducing generalization error caused by excessive neurons while the latter for sparsifying input signals in the presence of substantial redundancy. Unlike previous theoretical work, we consider the $L_1$ loss, which ranks among the most popular loss functions in, e.g., learning from ordinal data (Pedregosa et al., 2017) or imaging data (Zhao et al., 2016), and for which the statistical risk has not been studied previously. In practice, the use of $L_1$ loss for training has been implemented in prevalent computational frameworks such as Tensorflow (Google, 2016), Pytorch (Ketkar, 2017), and Keras (Gulli & Pal, 2017).

## 2 PROBLEM FORMULATION

### 2.1 MODEL ASSUMPTION AND EVALUATION

Suppose we have $n$ labeled observations $\{(x_i, y_i)\}_{i=1,\dots,n}$, where $y_i$'s are continuously-valued responses or labels. We assume that the underlying data generating model is $y_i = f_*(x_i) + \varepsilon_i$ for some unknown function $f_*(\cdot)$, where $x_i$'s $\in \mathbb{X} \subset \mathbb{R}^d$ are independent and identically distributed,

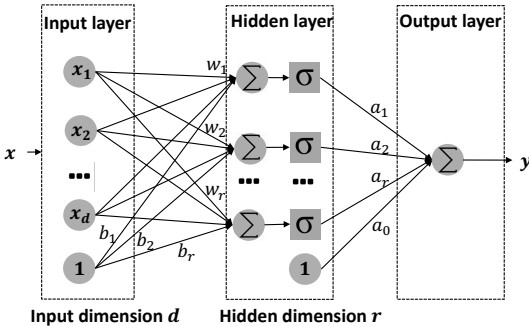

Figure 1: A graph showing the two-layer neural network model considered in (2).

and $\varepsilon_i$'s are independent and identically distributed that is symmetric at zero and

$$\mathbb{E}\left(\varepsilon_i^2 \mid x_i\right) \leq \tau^2. \tag{1}$$

Here, $\mathbb{X}$ is a bounded set that contains zero, for example $\{x : \|x\|_\infty \leq M\}$ for some constant $M$. Our goal is learn a regression model $\hat{f}_n : x \mapsto \hat{f}_n(x)$ for prediction. The $\hat{f}_n$ is obtained from the following form of neural networks

$$\sum_{j=1}^{r} a_j \sigma(w_j^\top x + b_j) + a_0, \tag{2}$$

where $a_0, a_j, b_j \in \mathbb{R}, w_j \in \mathbb{R}^d$, $j = 1, \ldots, r$, are parameters to estimate. We let $a = [a_0, a_1, \ldots, a_r]^\mathrm{T}$ denote the output layer coefficients. An illustration is given Figure 1. The estimation is typically accomplished by minimizing the empirical risk $n^{-1} \sum_{i=1}^{n} \ell(y_i, f(x_i))$, for some loss function $l(\cdot)$ plus a regularization term. We first consider the $L_1$ regularization at the output layer. In particular, we search for such $f$ by the empirical risk minimization from the function class

$$\mathcal{F}_V = \left\{ f : \mathbb{R}^d \to \mathbb{R} \middle| f(x) = \sum_{j=1}^{r} a_j \sigma(w_j^\top x + b_j) + a_0, \|a\|_1 \leq V \right\} \tag{3}$$

where $V$ is a constant. The following statistical risk measures the predictive performance of a learned model $f$:

$$\mathcal{R}(f) \triangleq \mathbb{E}\,\ell(y, f(x)) - \mathbb{E}\,\ell(y, f_*(x)).$$

The loss function $\ell(\cdot)$ is pre-determined by data analysts, usually the $L_1$ loss defined by $\ell(y, \tilde{y}) = |y - \tilde{y}|$ or the $L_2$ loss defined by $\ell_2(y, \tilde{y}) = (y - \tilde{y})^2$. Under the $L_1$ loss, the risk is $\mathcal{R}(f) = \mathbb{E}\,|f_*(x) + \varepsilon - f(x)| - \mathbb{E}\,|\varepsilon|$, which is nonnegative for symmetric random variables $\varepsilon$. It is typical to use the same loss function for both training and evaluation.

## 2.2 NOTATION

Throughout the paper, we use $n, d, k, r$ to denote the number of observations, the number of input variables or input dimension, the number of significant input variables or sparsity level, the number of neurons (or hidden dimension), respectively. We write $a_n \gtrsim b_n$, $b_n \lesssim a_n$, or $b_n = O(a_n)$, if $|b_n/a_n| < c$ for some constant $c$ for all sufficiently large $n$. We write $a_n \asymp b_n$ if $a_n \gtrsim b_n$ as well as $a_n \lesssim b_n$. Let $\mathcal{N}(\boldsymbol{\mu}, V)$ denote Gaussian distribution with mean $\boldsymbol{\mu}$ and covariance $V$. Let $\|\cdot\|_1$ and $\|\cdot\|_2$ denote the common $L_1$ and $L_2$ vector norms, respectively. Let $\mathbb{X}$ denote the essential support of $X$. For any vector $\boldsymbol{z} \in \mathbb{R}^d$, we define $\|\boldsymbol{z}\|_{\mathbb{X}} \triangleq \sup_{x \in \mathbb{X}} |x^\top \boldsymbol{z}|$, which may or may not be infinity. If $\mathbb{X} = \{x : \|x\|_\infty \leq M\}$, $\|\boldsymbol{z}\|_{\mathbb{X}}$ is equivalent to $M\|\boldsymbol{z}\|_1$. Throughout the paper, $\hat{f}_n$ denotes the estimated regression function with $n$ being the number of observations.

## 2.3 ASSUMPTIONS AND CLASSICAL RESULTS

We introduce some technical assumptions necessary for our analysis, and state-of-the-art statistical risk bounds built through dimension-based complexity analysis.

*Assumption* 1. The activation function $\sigma(\cdot)$ is a bounded function on the real line satisfying $\sigma(x) \to 1$ as $x \to \infty$ and $\sigma(x) \to 0$ as $x \to -\infty$, and it is $L$-Lipschitz for some constant $L$.

*Assumption* 2. The regularization constant $V$ is larger than $2C + f_*(0)$, where $C$ is any constant such that the Fourier transform of $f_*$, denoted by $F$, satisfies

$$\int_{\mathbb{R}^d} \|\omega\|_{\mathbb{X}} F(d\omega) \leq C. \tag{4}$$

*Assumption* 3. $\sigma(x)$ approaches its limits at least polynomially fast, meaning that $|\sigma(x) - \mathbf{1}\{x > 0\}| < \varepsilon$ for all $|x| > x_\varepsilon$ where $x_\varepsilon$ is a polynomial of $1/\varepsilon$. Also, the value of $\eta \overset{\triangle}{=} \sup_j \|w_j\|_{\mathbb{X}}$ scales with $n$ polynomially meaning that $\log \eta = O(\log n)$ as $n \to \infty$.

*Assumption* 4. There exists a constant $c > 0$ and a bounded subset $\mathcal{S} \subset \mathbb{R}$ such that $\mathbb{P}(X \in \mathcal{S}) > c$ and $\inf_{x \in \mathcal{S}} \sigma'(x) > c$ for $X \sim \mathcal{N}(0, 1)$.

We explain each assumption below. The above notation of $C, V$ follow those in (Barron, 1993; 1994). Assumption 1 specifies the class of the activation functions we consider. A specific case is the popular activation function $\sigma(x) = 1/\{1 + \exp(-x)\}$. Assumption 2, first introduced in (Barron, 1993), specifies the smoothness condition for $f_*$ to ensure the approximation property of neural networks (see Theorem 2.1). In Assumption 3, the condition for $w$ is for technical convenience. It could also be replaced with the following alternative condition: There exists a constant $c > 0$ such that the distribution of $x$ satisfies

$$\sup_{w:\|w\|_2=1} \mathbb{P}\big(\log(|w^\top x|) < c \log \varepsilon\big) < \varepsilon$$

for any $\varepsilon \in (0, 1)$. Simply speaking, the input data $x$ is not too small with high probability. This condition is rather mild. For example, it holds when each component of $x$ has a a bounded density function. This alternative condition ensures that for some small constant $\varepsilon > 0$ and any $w \in \mathbb{R}^d$, there exists a surrogate of $w$, $\hat{w} \in \mathbb{R}^d$ with $\log \|\hat{w}\|_2 = O(-\log \varepsilon)$, such that

$$\mathbb{P}(|\sigma(w^\top x) - \sigma(\hat{w}^\top x)| > \varepsilon) < \varepsilon.$$

And this can be used to surrogate the assumption of $w$ in Assumption 3 throughout the proofs in the appendix. Assumption 4 means that $\sigma(\cdot)$ is not a nearly-constant function. This condition is only used to bound the minimax lower bound in Theorem 3.2.

**Theorem 2.1** (Approximation error bound (Barron, 1993)). *Suppose that Assumptions 1, 2, 3 hold. We have*

$$\inf_{f \in \mathcal{F}_V} \left\{ \int_{\mathbb{X}} (f(x) - f_*(x))^2 \mu(dx) \right\}^{1/2} \leq 2C \left( \frac{1}{\sqrt{r}} + \delta_\eta \right),$$

*where $\mu$ denotes a probability measure on $\mathbb{X}$,*

$$\delta_\eta = \inf_{0 < \varepsilon < 1/2} \left\{ 2\varepsilon + \sup_{|x| > \varepsilon} |\sigma(\eta x) - \mathbf{1}\{x > 0\}| \right\}, \tag{5}$$

*$\eta$ is defined in Assumption 3, and $C$ is defined in (4).*

**Theorem 2.2** (Statistical risk bound (Barron, 1994)). *Suppose that Assumptions 1, 2, 3 hold. Then the $L_2$ estimator $\hat{f}_n$ in $\mathcal{F}_V$ satisfies $\mathbb{E}\{\hat{f}_n(x) - f_*(x)\}^2 \lesssim V^2/r + (rd \log n)/n$. In particular, if we choose $r \asymp V \sqrt{n/(d \log n)}$, then $\mathbb{E}\{\hat{f}_n(x) - f_*(x)\}^2 \lesssim V \sqrt{(d \log n)/n}$.*

It is known that a typical parametric rate under the $L_2$ loss is at the order of $O(d/n)$, much faster than the above result. This gap is mainly due to excessive model complexity in bounding generalization errors. We will show in Section 3 that the gap in the rate of convergence can be filled when using $L_1$ loss. Our technique will be based on the machinery of Rademacher complexity, and we bound this complexity through a joint analysis of the norm of coefficients ('norm-based') as well as dimension of parameters ('dimension-based').

## 2.4 MODEL COMPLEXITY AND GENERALIZATION ERROR

The statistical risk consists of two parts. The first part is an approximation error term non-increasing in the number of neurons $r$, and the second part describes generalization errors. The key issue for

risk analysis is to bound the second term using a suitable model complexity and then tradeoff with the first term. We will develop our theory based on the following measure of complexity.

Let $\mathcal{F}$ denote a class of functions each mapping from $\mathbb{X}$ to $\mathbb{R}$, and $x_1, x_2, \ldots, x_n \in \mathbb{X}$. Following a similar terminology as in (Neyshabur et al., 2015), the Rademacher complexity, or simply 'complexity', of a function class $\mathcal{F}$ is defined by $\mathbb{E} \sup_{f \in \mathcal{F}} |n^{-1} \sum_{i=1}^{n} \xi_i f(x_i)|$, where $\xi_i, i = 1, 2, \ldots, n$ are independent symmetric Bernoulli random variables.

**Lemma 2.3** (Rademacher complexity of $\mathcal{F}_V$). *Suppose that Assumptions 1, 3 hold. Then for the Rademacher complexity of $\mathcal{F}_V$, we have*

$$\mathbb{E} \sup_{f \in \mathcal{F}_V} \left| \frac{1}{n} \sum_{i=1}^{n} \xi_i f(x_i) \right| \lesssim \frac{V\sqrt{d \log n}}{\sqrt{n}}. \tag{6}$$

The proof is included in Appendix A.1. The bound in (6) is derived from an amalgamation of dimension-based and norm-based analysis elaborated in the appendix. It is somewhat surprising that the bound does not explicitly involve the approximation error part (that depends on $r$ and $\eta$). This Rademacher complexity bound enables us to derive tight statistical risk bounds in the following section.

## 3 MAIN RESULTS

### 3.1 STATISTICAL RISK BOUND FOR THE $L_1$ REGULARIZED NETWORKS IN (3)

**Theorem 3.1** (Statistical risk bound). *Suppose that Assumptions 1, 2, 3 hold. Then the constrained $L_1$ estimator $\hat{f}_n$ over $\mathcal{F}_V$ satisfies*

$$\mathcal{R}(\hat{f}_n) \lesssim \left( \frac{1}{\sqrt{r}} + \delta_\eta \right) C + \frac{V\sqrt{d \log n} + \tau}{\sqrt{n}}, \tag{7}$$

*where $\delta_\eta$ is defined in (5), and $\tau$ was introduced in (1). Moreover, choosing the parameters $r, \eta$ large enough, we have*

$$\mathcal{R}(\hat{f}_n) \lesssim \frac{V\sqrt{d \log n} + \tau}{\sqrt{n}}. \tag{8}$$

The proof is in Appendix A.2. We briefly explain our main idea in deriving the risk bound (7). A standard statistical risk bound contains two parts which correspond to the approximation error and generalization error, respectively. The approximation error part in (7) is the first term, which involves the hidden dimension $r$ and the norm of input coefficients through $\eta$. This observation motivates us to use the norm of output-layer coefficients through $V$ and the input dimension $d$ to derive a generalization error bound. In this way, the generalization error term does not involve $r$ already used for bounding the approximation error, and thus a bias-variance tradeoff through $r$ is avoided. This thought leads to the generalization error part in (7), which is the second term involving $V$ and $d$. Its proof combines the machinery of both dimension-based and norm-based complexity analysis. From our analysis, the error bound in Theorem 3.1 is a consequence of the $L_1$ loss function and the employed $L_1$ regularization. In comparison with the previous result of Theorem 2.2, the bound obtained in Theorem 3.1 is tight and it approaches the parametric rate $\sqrt{d/n}$ for the $d < n$ regime. Though we can only prove for $L_1$ loss in this work, we conjecture that the same rate is achieved using $L_2$ loss.

In the following, we further show that the above risk bound is minimax optimal. The minimax optimality indicates that deep neural networks with more than two layers will not perform much better than shallow neural networks when the underlying regression function belongs to $\mathcal{F}_V$.

**Theorem 3.2** (Minimax risk bound). *Suppose that Assumptions 1 and 4 hold, and $x_1, x_2, \ldots, x_n \overset{iid}{\sim} \mathcal{N}(0, \boldsymbol{I}_d)$, then $\inf_{\hat{f}_n} \sup_{f \in \mathcal{F}_V} \mathcal{R}(\hat{f}_n(x)) \gtrsim V\sqrt{d/n}$.*

Here the $\mathcal{F}_V$ is the same one as defined in (3). All the smooth functions $f_*(\cdot)$ that satisfy $V > 2C + f_*(0)$ and (4) belong to $\mathcal{F}_V$ according to Theorem 2.1. The proof is included in Appendix A.3.

### 3.2 ADAPTIVENESS TO THE INPUT SPARSITY

It is common to input a large dimensional signal to a neural network, while only few components are genuinely significant for prediction. For example, in environmental science, high dimensional weather signals are input for prediction while few are physically related (Shi et al., 2015). In image processing, the image label is relevant to few background pixels (Han et al., 2015). In natural language processing, a large number of redundant sentences sourced from Wikipedia articles are input for language prediction (Diao et al., 2019). The practice motivates our next results to provide a tight risk bound for neural networks whose input signals are highly sparse.

*Assumption* 5. There exists a positive integer $k \leq d$ and an index set $S \subset \{1, \ldots, d\}$ with $\mathrm{card}(S) = k$, such that $f_*(x) = g_*(x_S)$ for some function $g_*(\cdot)$ with probability one.

The subset $S$ is generally unknown to data analysts. Nevertheless, if we know $k$, named the sparsity level, the risk bound could be further improved by a suitable regularization on the input coefficients. We have the following result where $d$ is replaced with $k$ in the risk bound of Theorem 3.1.

*Proposition* 3.3. Suppose that that Assumptions 1, 2, 3, 5 hold. Suppose that $\hat{f}_n$ is the $L_1$ estimator over the following function class

$$\left\{ f : \mathbb{R}^d \to \mathbb{R} \Big| f(x) = \sum_{j=1}^r a_j \sigma(w_j^\top x + b_j) + a_0, \|a\|_1 \leq V, \sup_j \|w_j\|_0 \leq k \right\}.$$

Then $\mathcal{R}(\hat{f}_n) \lesssim \sqrt{\{k \log(dn)\}/n}$.

The proof is included in Appendix A.4. The above statistical risk bound is also minimax optimal according to a similar argument in Theorem 3.2. From a practical point of view, the above $L_0$ constraint is usually difficult to implement, especially for a large input dimension $d$. Alternatively, one may impose an $L_1$ constraint instead of an $L_0$ constraint on the input coefficients. Our next result is concerned with the risk bound when the model is learned from a joint regularization on the output and input layers. For technical convenience, we will assume that $\mathbb{X}$ is a bounded set.

**Theorem 3.4.** *Consider the following function class of two-layer neural networks*

$$\mathcal{F}_{V,\eta} = \left\{ f : \mathbb{R}^d \to \mathbb{R} \Big| f(x) = \sum_{j=1}^r a_j \sigma(w_j^\top x + b_j) + a_0, \|a\|_1 \leq V, \sup_{1 \leq j \leq r} (\|w_j\|_1 + |b_j|) \leq \eta \right\}.$$

*Suppose that $V \gtrsim C$, where $C$ is defined in (4). Then the constrained $L_1$ estimator $\hat{f}_n$ over $\mathcal{F}_{V,\eta}$ satisfies*

$$\mathcal{R}(\hat{f}_n) \lesssim C \left( \frac{1}{\sqrt{r}} + \delta_\eta \right) + \frac{V\eta + \tau}{\sqrt{n}},$$

*where $\delta_\eta$ is defined in (5). In particular, choosing $r$ large enough, we have*

$$\mathcal{R}(\hat{f}_n) \lesssim C\delta_\eta + \frac{V\eta + \tau}{\sqrt{n}}$$

*which does not involve the input dimension $d$ and the number of hidden neurons $r$. Moreover, suppose that $\sigma(x) = 1/(1 + e^{-x})$, $\eta \asymp \left( n \log^2 n \right)^{1/3}$, then $\mathcal{R}(\hat{f}_n) \lesssim V \{(\log n)/n\}^{1/3}$.*

The proof is included in Appendix A.5. In the above result, the risk bound is at the order of $O(n^{-1/3})$, which is slower than the $O(n^{-1/2})$ in the previous Theorem 3.1 and Proposition 3.3 if ignoring $d$ and logarithmic factors of $n$. However, for a large input dimension $d$ that is even much larger than $n$, the bound can be much tighter than the previous bounds since it is dimension-free.

## 4 CONCLUSION AND FURTHER REMARKS

We studied the tradeoff between model complexity and statistical risk in two-layer neural networks from the explicit regularization perspective. We end our paper with two future problems. First, in Theorem 3.4, For a small $d$, the order of $n^{-1/3}$ seems to be an artifact resulting from our technical arguments. We conjecture that in the small $d$ regime, this risk bound could be improved to $O(n^{-1/2})$ by certain adaptive regularizations. Second, it would be interesting to emulate the current approach to yield similarly tight risk bounds for deep forward neural networks.

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

# A  APPENDIX

## A.1  PROOF OF LEMMA 2.3

We first prove (6), which uses an amalgamation of dimension-based and norm-based analysis. For the output layer, we use the following norm-based analysis

$$\mathbb{E} \sup_{f \in \mathcal{F}_V} \left| \frac{1}{n} \sum_{i=1}^{n} \xi_i f(\boldsymbol{z}_i) \right| = \mathbb{E} \sup_{f \in \mathcal{F}_V} |\langle a, \frac{1}{n} \sum_{i=1}^{n} \xi_i \sigma(\boldsymbol{W}^\top \boldsymbol{z}_i + \boldsymbol{b}) \rangle| \tag{9}$$

$$\leq \sup \|a\|_1 \mathbb{E} \sup_{f \in \mathcal{F}_V} \left\| \frac{1}{n} \sum_{i=1}^{n} \xi_i \sigma(\boldsymbol{W}^\top \boldsymbol{z}_i + \boldsymbol{b}) \right\|_\infty \leq V \mathbb{E} \sup_{f \in \mathcal{F}_V} \max_j \left| \frac{1}{n} \sum_{i=1}^{n} \xi_i \sigma(w_j^\top \boldsymbol{z}_i + b_j) \right|$$

$$\leq V \mathbb{E} \sup_{w \in \mathbb{R}^d} \left| \frac{1}{n} \sum_{i=1}^{n} \xi_i \sigma(w^\top \boldsymbol{z}_i + b) \right|.$$

For notational convenience, we define $w_0 = 0, b_0 = 0$, and $a_0 = \sigma(0)^{-1} a_0 \sigma(w_0^\top \boldsymbol{z} + b_0)$ so that $a_0$ can be treated in a similar manner as other $a_i$'s. Without loss of generality, we do not separately consider $a_0$ in the following proofs.

Next, we prove that

$$\mathbb{E} \sup_{w \in \mathbb{R}^d} \left| \frac{1}{n} \sum_{i=1}^{n} \xi_i \sigma(w^\top \boldsymbol{z}_i + b) \right| \lesssim \sqrt{\frac{d \log n}{n}}, \tag{10}$$

and thus conclude the proof. The proof will be based on an $\varepsilon$-net argument together with the union bound. For any $\varepsilon$, let $W_\varepsilon \subset \mathbb{R}^d$ denote the subset

$$W_\varepsilon = \left\{ w = \frac{\varepsilon}{2d}(i_1, i_2, \ldots, i_d) : i_j \in \mathbb{Z}, \|w\|_1 \leq \eta_n \right\}.$$

Then, for any $w, b$, there exists some element $\hat{w} \in W_\varepsilon$ such that

$$\sup_{\boldsymbol{z} \in \mathbb{X}} |\sigma(w^\top \boldsymbol{z} + b) - \sigma(\hat{w}^\top \boldsymbol{z} + \hat{b})| \leq \sup_{\boldsymbol{z}} |(w^\top \boldsymbol{z} + b) - (\hat{w}^\top \boldsymbol{z} + \hat{b})| \leq \sup_{\boldsymbol{z}} |(w - \hat{w})^\top \boldsymbol{z}| + |b - \hat{b}|$$

$$\leq \|w - \hat{w}\|_1 \sup_{\boldsymbol{z}} \|\boldsymbol{z}\|_\infty + |b - \hat{b}| \leq \varepsilon,$$

where $\hat{b} = (\varepsilon/2d) \lfloor (2db/\varepsilon) \rfloor$ and $\lfloor \cdot \rfloor$ is the floor function. By Bernstein's Inequality, for any $w, b$,

$$\mathbb{P}\left(|\frac{1}{n}\sum_{i=1}^{n}\xi_i\sigma(w^\top z_i + b)| > t\right) \le 2\exp\left\{-\frac{nt^2}{2(1+t/3)}\right\}.$$

By taking the union bound over $W_\varepsilon$, and use the fact that $\log \text{card}(W_\varepsilon) \lesssim d\log(nd/\varepsilon)$, we obtain

$$\sup_{w\in\mathbb{R}^d}\left|\frac{1}{n}\sum_{i=1}^{n}\xi_i\sigma(w^\top z_i + b)\right| \lesssim \varepsilon + \sqrt{\frac{d}{n}\log\frac{nd}{\varepsilon}\log\frac{1}{\delta}},$$

with probability at least $1-\delta$. Then the desired result is obtained by taking $\varepsilon \sim \sqrt{(d\log n)/n}$.

## A.2 PROOF OF THEOREM 3.1

The proof is based on the following contraction lemma used in (Neyshabur et al., 2015).

**Lemma A.1** (Contraction Lemma). *Suppose that $g$ is $L$-Lipschitz and $g(0) = 0$. Then for any function class $\mathcal{F}$ mapping from $\mathbb{X}$ to $\mathbb{R}$ and any set $\{x_1, x_2, \ldots, x_n\}$, we have*

$$\mathbb{E}\sup_{f\in\mathcal{F}}\left|\frac{1}{n}\sum_{i=1}^{n}\xi_i g(f(x_i))\right| \le 2L\mathbb{E}\sup_{f\in\mathcal{F}}\left|\frac{1}{n}\sum_{i=1}^{n}\xi_i f(x_i)\right|. \tag{11}$$

With the above lemma, we have the following result.

**Lemma A.2.** *The constrained $L_1$ estimator $\hat{f}_n$ over $\mathcal{F}$ satisfies*

$$\mathcal{R}(\hat{f}_n) \le \min_{f\in\mathcal{F}}\mathbb{E}|f(x) - f_*(x)| + 2\mathbb{E}\sup_{f\in\mathcal{F}}|\frac{1}{n}\sum_{i=1}^{n}\xi_i f(z_i)| + 2\sqrt{\frac{\mathbb{E}y^2}{n}}. \tag{12}$$

*Proof.* Define the empirical risk as:

$$\mathcal{R}_n(f) = \mathbb{E}\left(\frac{1}{n}\sum_{i=1}^{n}|f_*(x_i) + \varepsilon_i - f(x_i)|\right) - \mathbb{E}|\varepsilon|. \tag{13}$$

Since $\hat{f}_n$ minimizes $n^{-1}\sum_{i=1}^{n}|f_*(x_i) + \varepsilon_i - f(x_i)|$ in $\mathcal{F}$, we have

$$\mathcal{R}(\hat{f}_n) \le \mathcal{R}(\hat{f}_n) - \{\mathcal{R}_n(\hat{f}_n) - \mathcal{R}_n(\hat{f})\} = \{\mathcal{R}(\hat{f}_n) - \mathcal{R}_n(\hat{f}_n)\} + \mathcal{R}_n(f_0), \tag{14}$$

where $f_0 = \arg\min_{f\in\mathcal{F}}\mathcal{R}(f)$. We also have

$$\mathcal{R}_n(f_0) = \mathcal{R}(f_0) = \min_{f\in\mathcal{F}}\mathbb{E}(|f_*(x) + \varepsilon - f(x_i)| - |\varepsilon|) \le \min_{f\in\mathcal{F}}\mathbb{E}|f(x) - f_*(x)|. \tag{15}$$

In the following, we will analyze the term $\mathcal{R}(\hat{f}_n) - \mathcal{R}_n(\hat{f}_n)$ in (14). Let $z_i$'s denote independent and identically distributed copies of $x_i$'s.

$$\mathcal{R}(\hat{f}_n) - \mathcal{R}_n(\hat{f}_n) = \mathbb{E}\frac{1}{n}\sum_{i=1}^{n}\left\{|\hat{f}_n(z_i) - f_*(z_i) - \varepsilon_i| - |\hat{f}_n(x_i) - f_*(x_i) - \varepsilon_i|\right\}$$

$$\le \mathbb{E}\sup_{f\in\mathcal{F}}\frac{1}{n}\sum_{i=1}^{n}\left\{|f(z_i) - f_*(z_i) - \varepsilon_i| - |f(x_i) - f_*(x_i) - \varepsilon_i|\right\}$$

$$\le 2\mathbb{E}\sup_{f\in\mathcal{F}}\frac{1}{n}\sum_{i=1}^{n}\xi_i|f(z_i) - f_*(z_i) - \varepsilon_i|,$$

where $\xi_1, \ldots, \xi_n$ are independent and identically distributed symmetric Bernoulli random variables that are independent with $z_i$'s. According to Lemma A.1, since $g(x) = |x|$ is 1-Lipschitz and $g(0) = 0$, we have

$$\mathbb{E}\sup_{f\in\mathcal{F}}\frac{1}{n}\sum_{i=1}^{n}\xi_i|f(z_i) - f_*(z_i) - \varepsilon_i| \le 2\mathbb{E}\sup_{f\in\mathcal{F}}|\frac{1}{n}\sum_{i=1}^{n}\xi_i(f(z_i) - f_*(z_i) - \varepsilon_i)|$$

$$\le 2\mathbb{E}\sup_{f\in\mathcal{F}}\left|\frac{1}{n}\sum_{i=1}^{n}\xi_i f(z_i)\right| + 2\sqrt{\frac{\mathbb{E}y^2}{n}}.$$

Combining this and (15), we conclude the proof of Lemma A.2. $\qquad\square$

**Proof of Theorem 3.1**. The proof of (7) is a direct consequence of Lemma 2.3, Lemma A.2, Theorem 2.1 and the fact that the first moment is no more than the second moment. The proof of (8) follows from the fact that $\delta(\eta) \to 0$ as $\eta \to \infty$.

## A.3 PROOF OF THEOREM 3.2

Define a subclass of $\mathcal{F}_V$ by

$$\mathcal{F}_0 = \left\{ f : \mathbb{R}^d \to \mathbb{R} \,\middle|\, f(x) = V\sigma(w^\top x), \|w\|_2 = 1 \right\}.$$

In the following, we will prove the minimax bound for $\mathcal{F}_V$ by analyzing $\mathcal{F}_0$. Notice that

$$\mathbb{E}\,|\sigma(w_1^\top x) - \sigma(w_2^\top x)| \geq \mathbb{E}\,\inf_u \sigma'(u) \cdot |w_1^\top x - w_2^\top x| \cdot \mathbb{I}(w_1^\top x, w_2^\top x \in \mathcal{S}) \gtrsim \|w_1 - w_2\|_2.$$

Let $M_1(\varepsilon)$ denote the packing $\varepsilon$-entropy of $\mathcal{F}_0$ with $L_1$ distance, then $M_1(\varepsilon)$ is greater than the packing $\varepsilon$-entropy of $\mathbb{B}_1^d$ with $L_2$ distance, which means $M_1(\varepsilon) \gtrsim d$. Let $V_k(\varepsilon)$ denote the covering $\varepsilon$-entropy of $\mathcal{F}_0$ with the square root Kullback-Leibler divergence, then according to its relation with the $L_2$ distance shown in (Yang & Barron, 1999), we have

$$V_k(\varepsilon) \leq M_2(\sqrt{2}\varepsilon) \lesssim d\log\frac{1}{\varepsilon},$$

where $M_2(\varepsilon)$ denote the packing $\varepsilon$-entropy of $\mathcal{F}_V$ with $L_2$ loss function. The second inequality is proved in a similar way to the proof of Lemma 2.3, which is omitted here for brevity. Hence, according to (Yang & Barron, 1999, Theorem 1),

$$\inf_{\hat{f}_n} \sup_{f \in \mathcal{F}_V} \mathcal{R}(\hat{f}_n(x)) \geq \inf_{\hat{f}_n} \sup_{f \in \mathcal{F}_0} \mathcal{R}(\hat{f}_n(x)) \gtrsim V\sqrt{\frac{d}{n}},$$

This concludes the proof.

## A.4 PROOF OF PROPOSITION 3.3

To prove the proposition, it is sufficient to verify the following Rademacher complexity bound

$$\mathbb{E}\,\sup\left|\frac{1}{n}\sum_{i=1}^n \xi_i\sigma(w^\top z_i + b)\right| \lesssim \sqrt{k\log d\log n},$$

which can be derived easily by adjusting the proof in Lemma 2.3. Then the result follows with a similar analysis as in Theorem 3.1.

## A.5 PROOF OF THEOREM 3.4

It can be verified from the identity (9) that

$$\mathbb{E}\,\sup_{f \in \mathcal{F}_V}\left|\frac{1}{n}\sum_{i=1}^n \xi_i f(x_i)\right| \leq \sum_{j=0}^r \mathbb{E}\,\sup_{f \in \mathcal{F}_V}|a_j|\left|\frac{1}{n}\sum_{i=1}^n \xi_i\sigma(w_j^\top x_i + b_j)\right|. \tag{16}$$

Then according to Lemma A.1, we have

$$\mathbb{E}\,\sup_{f \in \mathcal{F}_V}\left|\frac{1}{n}\sum_{i=1}^n \xi_i\sigma(w_j^\top x_i + b_j)\right| \lesssim \sqrt{\frac{\log n}{n}}(\|w_j\|_{\mathbb{X}} + |b_j|). \tag{17}$$

Combining (16) and (17), we obtain the following lemma that may be interesting on its own right.

**Lemma A.3.** *We have*

$$\mathbb{E}\,\sup_{f \in \mathcal{F}_V}\left|\frac{1}{n}\sum_{i=1}^n \xi_i f(x_i)\right| \lesssim \sqrt{\frac{\log n}{n}}\sum_{j=0}^r |a_j|(\|w_j\|_{\mathbb{X}} + |b_j|) \lesssim V\sqrt{\frac{\log n}{n}}\max_j \|w_j\|_{\mathbb{X}}.$$

Since $\|w\|_{\mathbb{X}} \lesssim \|w\|_1$ and $\{w : \|w\|_{\mathbb{X}} \lesssim \eta\} \subset \{w : \|w\|_1 \lesssim \eta\}$, the $\|\cdot\|_{\mathbb{X}}$ can be replaced with $\|\cdot\|_1$ in the bounds in Lemmas A.3 and A.2. Then, with a similar argument as in the proof of Theorem 3.1, we conclude the proof of Theorem 3.4.

