# OpenReview forum: "THE EFFICACY OF L1 REGULARIZATION IN NEURAL NETWORKS"
_ICLR.cc/2021/Conference — Reject_

### Official Review · AnonReviewer2 · 2020-10-24
**Analysis seems standard and nor rigorous**

**Rating:** 5
**Confidence:** 4

**Review:**

This paper studies the statistical risk bounds for two-layer neural networks with $L_1$-regularization. The authors consider two types of $L_1$-regularization: the $L_1$-regularization on output layer and the $L_1$-regularization on the input layer. For the $L_1$-regularization on output layer, the authors develop nearly minimax statistical risk bounds. For the $L_1$-regularization on input layers, the authors develop bounds with no-dependency on the input dimension. The paper is clearly written and easy to follow.

Comments

1. This is a theory paper. However, as far as I can see, theoretical analysis is a bit standard. It seems that the authors do not introduce new techniques and idea in statistical learning theory (SLT). Furthermore, the results are a bit standard. For example, the statistical risk bound of the order $\sqrt{d/n}$ in Thm 3.1 is standard in SLT for general machine learning models. It is not surprising that this holds for two-layer neural networks.

2. In my opinion, eq (10) is not correct. Since there is no constraints on $w$, the left-hand side can be infinity by taking $w$ of infinite magnitude. Therefore, I think one should also imposes a constraint on $w$ in the definition of $F_v$ in eq (13)

3. In the definition of $F_{v,\eta}$ in Thm 3.4, there is a constraint on $w$. I think this constraint may affect the approximation error established in Thm 2.1. Then this can further affect the statistical risk in Thm 3.4.

4. Eq (17) should be not correct. Indeed, the left-hand side depends on a function class, while the right-hand side depends on $w_j$ and $b_j$. This is not meaningful.

5. In eq (14), $\hat{f}$ should be $f_0$

---

> ### Author Response · Authors · 2020-11-17
> **Thank you for your comments. We address each concern below.**
>
> Thank you for your comments. We address each concern below.
>
> On contribution:
> We agree that the theoretical tool is not beyond the existing statistical learning theory. However, the discovery of the tight statistical risk bound (at the rate of n^{1/2} under L1 loss) is new. The existing tight bound for neural networks is n^{1/4} under L2 loss (equivalently, the square-root of MSE).
>
> On eq(10):
> There was a typo under the sup in (10). The sup should be over the w whose l1 norm is bounded by eta_n (which we introduced in Assumption 3).
>
> On constraint of w:
> The constraint on w in Thm 3.4 will not affect the approximation error established in Thm 2.1, according to the definition of (5) in Theorem 2.1.
>
> On Eq (17):
> We double checked that (17) is correct. The function class at the left-hand side does not show on the right side, because we applied the contraction lemma (Lemma A.1) in conjunction with norm inequalites.
>
> On Eq (14):
> We will fix this typo.

---

### Official Review · AnonReviewer4 · 2020-10-28
**Minor contribution**

**Rating:** 4
**Confidence:** 4

**Review:**

This paper studies generalization bounds for neural networks with the following kind of setup:
(0) 1 hidden layer and sigmoid-like activations. the weights in the input layer are bounded in either a general norm, or sometimes specifically the $\ell_1$-norm.
(1) the loss function is the L1 loss |y - \hat{y}|, as opposed to the more common square loss.
(2) the weights of the last layer of the network are bounded in the vector $\ell_1$-norm. They also consider the case where the input layer is bounded in Sec 3.2.

A small notational point: the authors use L1 to denote both cases, i.e. write L1 instead of $\ell_1$ to denote the vector $\ell_1$-norm
of the weights (i.e. the sum of absolute values). Sometimes I do see the vector norm written L1 anyway, as in this paper. However, the vector norm is more commonly written as $\ell_1$, and I think this would be a good idea to minimize confusion between (1) and (2) above, which are very different things. (E.g. In the linear regression world, the $\ell_1$-norm is associated with sparsity and $L_1$ loss is associated with robustness to outliers.)

The main result of this paper is a generalization bound for this class of neural networks, which comes down to technical Lemma 2.3 bounding the Rademacher complexity of the class. This result follows in a relatively straightforward way, because L1 loss is lipschitz (so we can use contraction lemma) and because all of the weights are bounded in $\ell_1$ norm; a small twist is they use that the input dimension is small to get a better bound on the term coming from the first layer of the network. They also state a result given by combining this generalization bound with Barron's approximation theorem.

It would be useful for the authors to compare their results further with previous work in this area. In particular, the idea of looking at the $\ell_1$ norm of the output layer for generalization bounds has been considered as part of the paper "The Sample Complexity of Pattern Classification with Neural Networks: The Size of the Weights is More Important than the Size of the Network", Bartlett '98. See e.g. Theorem 17.

Overall, this paper doesn't seem to contribute many fresh ideas to the study of generalization bounds for neural networks and so I would tend towards rejection.

minor notes:
- contraction lemma & rademacher complexity are basic results in this area, so preferably they shouldn't be cited from neyshabur et al (2015). For example, you could cite a textbook in the area, like Shalev-Shwartz + Ben-David book.

---

> ### Author Response · Authors · 2020-11-17
> **Thank you for your comments. We address each comment below.**
>
> Thank you for your comments. We address each comment below.
>
> Notational point:
> We will change the notation as you suggest.
>
> On related work:
> We will add a discussion on the related work that used norm-based analysis, including the mentioned work.
>
> On contraction lemma:
> We will cite earlier work as you suggest.

---

### Official Review · AnonReviewer3 · 2020-10-28
**Clearly written paper with interesting findings, but I'm concerned the contributions are few and limited in scope.**

**Rating:** 5
**Confidence:** 3

**Review:**

1. Summarize what the paper claims to contribute.

The paper bounds the excess L1 risk of two-layer neural networks when L1 regularization is applied to the second layer. The rates obtained are better than the rates with L2 risk and nearly optimal. Crucially, the L1 regularization removes the size of the hidden layer from the estimation error.

2. List strong and weak points of the paper.

Pros:
* The lower and upper bounds in the paper are almost matching.
* The discussion of the assumptions and results are clear and informative. I'm quite happy about this.
* The paper is overall clearly written and correctly executed (modulo minor issues that can be fixed).

Cons:
* The contributions have limited impact.

3. Clearly state your recommendation (accept or reject) with one or two key reasons for this choice.

Unfortunately I am recommending rejection. The paper is well written and well executed, but I am concerned that the contributions are few and that their impact is limited.

4. Provide supporting arguments for your recommendation.

Concerns:
* Limiting the setting to two-layer neural networks and excluding common activations like ReLU limits the impact of the results.
* The paper applies existing results/analyses to a new setting (combining neural networks and L1 regularization). This setting does not pose any particular technical challenge. The findings are interesting but few, in my opinion. It is unclear whether they are enough for acceptance.

5. Ask questions you would like answered by the authors to help you clarify your understanding of the paper and provide the additional evidence you need to be confident in your assessment.

Important questions/requests:
* Please clarify the significance of the contributions, and their potential impact.
* If the papers address a new technical challenge, please clarify.
* Are they widely applicable as they are?
* Has the paper opened up a path for future research to use the knowledge contributed, and to make the results more broadly applicable?

Tangential questions:
* Assumption 2 seems to include the assumption "our choice of V is appropriate". So one does not need to do model selection on V. The results remove r and introduce V into the estimation error, but is V easier to choose than r?

6. Provide additional feedback with the aim to improve the paper.

The paper uses Bernoulli random variables for the Rademacher complexity, but they should be Rademacher random variables (see Appendix A in Neyshabur et al., 2015).

(5) makes me wonder whether most of the error happens around zero, and whether easy cases are significantly easier

7. Typos/Minor questions

For eta in Assumption 3, should there be a sup over w?
In the definition of eta, maybe write \log \eta \asymp \log n ?

What's the randomness inside the P in the alternative definition of Assumption 3? I'm assuming it's X, based on the explanation that follows the equation.

The quantifiers in Thm 3.2 look strange, but I think it's because in the original result the \hat{\theta} (mapped to \hat{f}_n here) refers to the estimator and not the actual empirical minimizer. Could you please clarify the notation here?

For Assumption 5, I was wondering if you wouldn't want a sparse latent representation instead? I think the case where the inputs are actually sparse is not usual, but the case where you can learn a sparse representation at least from an empirical perspective seems more realistic.

p.8 typo in definition of a_0

Sigma's Lipschitz constant is missing from the derivation (first step to second) at the bottom of p.8. The derivations following that are missing some constants. Also, please cite sources for Bernstein's Inequality and the covering number of W_eps.

(14): \hat{f} -> f_0

In the equations following (15) for that derivation it would be useful to add how the second term (with y^2) came to be.

---

> ### Author Response · Authors · 2020-11-17
> **Thank you for your comments. We address each question below.**
>
> Thank you for your comments. We address each question below.
>
> On Assumption 2:
> In practice, for the purpose of model selection, we do need to search V within the regularized training. Theoretically, our contribution is to show tight statistical risk bounds (as dependent on the sample size n) for any fixed V. It is a different theoretical perspective compared with the counterpart analysis based on r.
>
> On Bernoulli random variables:
> It is standard to use the Bernoulli random variables to define the Rademacher complexity and Rademacher process (originally from the symmetrization method).
>
> On (5):
> The quantity in equation (5) was introduced for technical derivations. It does not imply most of the errors occur around zero.
>
> On eta in Assumption 3:
> There is no additional sup over w. The sup is already over j, which indexes the w.
> We do not require log eta to be at the same order of log n.
>
> On the randomness inside P:
> The randomness is from the random x (the input).
>
> The quantifiers in Thm 3.2:
> It is a minimax notaton, meant for all possible \hat{f}_n (based on data). We will clarify this in the revision.
>
> On Assumption 5:
> What you proposed is a very interesting idea. We conjecture that it is possible to extend the results to a sparse latent representation (not from the input layer). We will remark it as future work.
>
> On \hat{f} -> f_0:
> We will fix this typo
>
> On the second term (with y^2):
> Would you please let us know where it is? We could not locate the term with y^2 after (15).

---

### Public Comment · ~Johannes_Lederer1 · 2020-11-14
**Reference**

The authors might have missed our recent paper https://arxiv.org/abs/2006.00294 .

---

### Decision · Program_Chairs · 2021-01-07
**Final Decision**

**Decision:**

Reject

**Comment:**

The paper presents a generalization bounds for l1 regularized networks.  The reviewers thought the results were clear and sound, but on the other hand rely on rather standard technical tools, and their impact is limited.
One question is why this particular regularization is related to practical learning of neural nets where explicit regularization is not used. The authors may want to relate their results to e.g., Theorem 1 in https://arxiv.org/pdf/1412.6614.pdf.